# A Novel Technique for Shortening Orthodontic Treatment: The “JET System”

**DOI:** 10.3390/medicina58020150

**Published:** 2022-01-19

**Authors:** Shinichi Narita, Kiyoko Narita, Masaru Yamaguchi

**Affiliations:** 1Jiyugaoka Orthodontic Clinic, Ono Building 3F, 5-24-10 Okuzawa, Setagaya-ku, Tokyo 158-0083, Japan; yamaguchimasaru11@gmail.com; 2Ginza Orthodontic Clinic, Ginza Granvia 6F, 3-3-14 Ginza, Chuo-ku, Tokyo 104-0061, Japan; yamaguchimasaru3@gmail.com

**Keywords:** regional acceleratory phenomenon, light continuous forces, low friction, orthodontic tooth movement, myofunctional therapy

## Abstract

We have developed a novel technique, the Jiyugaoka Enjoyable Treatment (JET) system, to complete orthodontic treatment in a short time. It entails the use of the regional acceleratory phenomenon (RAP), light continuous forces and low friction in cases involving extraction. In the JET system, tooth extraction not only creates space, but also triggers the RAP; thus tooth extraction accelerates orthodontic treatment. We describe for the first time how to use the JET system to shorten treatment time in a patient in whom four premolars were extracted. A 15 year old girl patient exhibited an Angle Class I bimaxillary protrusion with moderate crowding in the maxillary (−5.0 mm) and mandibular arches (−3.5 mm). Her facial appearance was slightly asymmetric, and her facial profile was convex. Immediately after the simultaneous extraction of the maxillary first premolars and mandibular second premolars, orthodontic treatment was started with a combination of passive self-ligating brackets and super-elastic nickel-titanium closed coil springs that provided orthodontic forces of less than 50 gf (1.8 ozf). The appliance was adjusted once a month. The total treatment time was 13 months. Cephalometric superimpositions showed a slight anchorage loss, and panoramic radiographs showed a slight apical root resorption but no significant reduction in the crest bone height. At the 3-year 6-month retention follow-up, stability was excellent. The JET system might shorten the orthodontic treatment period without serious anchorage loss or other adverse effects.

## 1. Introduction

In general, orthodontic treatment takes a long time. In a report on extraction cases, Yamazaki et al. [1] showed that the mean length of time for conventional active orthodontic treatment was 29 months. Most patients, however, want their treatment to be completed as early as possible.

Factors in the duration of treatment included the patient’s cooperation [2,3,4,5], malocclusion severity [3,4,5], presence or absence extractions [3,4,5,6,7], techniques involving the use of fixed appliances [5,8], and bracket prescription [9].

Many Japanese people have bimaxillary protrusion. In addition, when the maxillary canine teeth are in an abnormally high labial location, the dental arch in the maxilla is narrower [10], and extraction of other teeth is often needed.

The use of self-ligating (SL) brackets and low friction (LF) ligating brackets [11] has long been proposed as a way to release the frictional resistances that control orthodontic tooth movement at the bracket-archwire-ligature interface. Burrow [12] reported that SL brackets do not shorten treatment time because the effect of binding and notching, not friction, is significant, and the binding-and-releasing phenomenon occurs with SL brackets as well as with conventional (CV) brackets. SL brackets in which the self-ligating clip does not press against the archwire are called passive self-ligating (PSL) brackets [13]. PSL brackets have consistently created a smaller amount of friction than CV brackets have [14]. Therefore, PSL brackets may influence treatment duration.

We utilize super-elastic nickel-titanium (Ni-Ti) wires [15] and Ni-Ti closed coils [16] to generate light continuous forces (LCF). Yee et al. [17] reported that an LCF of 50 gf (1.8 ozf) provided a greater percentage of canine retraction than did heavy forces (300 gf), with less strain on the anchorage in humans. In rats, Tomizuka et al. [18] found that the LCF initially induced tooth movement better.

The system that combines these advantages (an LF and the LCF) was introduced by Damon [19]. The Damon system enables greater patient comfort during treatment, fewer visits to the orthodontists, a shorter overall treatment time, less need for extractions, and better outcomes in terms of both occlusal and facial esthetics. However, Fleming et al. [20] and Chen et al. [21] reported that the use of SL brackets did not reduce the overall treatment time or the total number of visits and did not result in a better occlusal outcome in comparison with CV brackets. Therefore, whether the Damon system shortens the overall treatment time is unclear.

Until now, researchers have denied that teeth move faster with PSL brackets. In previous studies [22,23], comparisons were made at a time when structural differences in the brackets did not affect treatment duration.

In order to use sliding mechanics [24] with the straight wire method, a rigid stainless steel (SS) wire was needed. Because of the limitation of these mechanics, there are few studies on shortening the treatment period by distally moving the canine teeth immediately after initiating treatment with a particular focus on the regional acceleratory phenomenon (RAP) by extraction. The RAP is a tissue reaction to a noxious stimulus that increases the healing ability of the affected tissues.

Now, however, instead of actively moving the canine tooth a few months after the brackets were attached, we used the Jiyugaoka Enjoyable Treatment (JET) system, which allows distal movement of the canine tooth with a 50 gf (1.8 ozf) Ni-Ti closed coil immediately after attachment and has fewer restrictions on tooth movement, thereby producing different results.

The JET system was developed to shorten the orthodontic treatment time for cases using the RAP by tooth extraction. The features and concepts of the JET system are illustrated in Figure 1.

## 2. Concepts of JET System

### 2.1. Zero-Step Method

In a conventional orthodontic treatment for extraction, a step-by-step treatment [25] is used to ensure reliable and high-quality results. A conventional treatment involves either three steps (alignment and leveling [step 1], space closure [step 2], anterior retraction [step 3]) or four steps (alignment and leveling [step 1], canine retraction [step 2], anterior retraction [step 3], and finishing [step 4]). In general, the treatment period is longer for cases involving extraction [3,4,5,6,7] because closing the space takes extra time. In addition, it is difficult to shorten the average orthodontic treatment time in extraction cases [20,21].

Therefore, to shorten the duration of orthodontic treatment while maintaining the quality of orthodontic treatment, we invented a method in which all steps start at the same time (Figure 1), including the space closure, instead of shortening each step, as in the conventional method.

### 2.2. Simultaneous Tooth Extraction

In the conventional orthodontic system, approximately six months elapse between tooth extraction and the initiation of the canine distal movement. This means that the distal movement of canine teeth and anterior retraction are initiated after the remodeling of the socket is completed. In the JET system, the aim is to finish the movement of most teeth before socket remodeling is completed; therefore, all the teeth to be extracted for orthodontic treatment are extracted on the same day, and the orthodontic appliance is placed on the following day to immediately initiate aggressive distal movement of the canine teeth.

Orthodontic tooth movement is a multi-step biological process characterized by sequential reactions of the periodontal tissue against biomechanical forces. The recruitment of osteoclast and osteoblast progenitor cells and the balanced activation of these cells around and within the periodontal ligament are essential for alveolar bone remodeling. Moreover, orthodontic tooth movement is reportedly related to inflammation. Osteoclasts and osteoblasts are increased by the RAP after fractures and surgery such as osteotomies and bone grafting, and bone healing is accelerated [26]. When a surgical incision was made into the head of the tibia in rabbits, new bone formed even in the trabecular bone around the incision area via an increased bone turnover [27,28]. In cases involving tooth extraction, extraction is followed immediately by the movement of the other teeth through the metabolic activity in the socket of the extracted tooth. This technique is based on a mechanism similar to the stimulation of tooth movement with corticotomy. Therefore, we use the JET system to perform canine retraction immediately after extraction of the premolars. In the JET system, tooth extraction not only creates space but also triggers the RAP; therefore, simultaneous tooth extraction accelerates orthodontic treatment.

### 2.3. Passive Self-Ligating Brackets (PSL)

Many orthodontists still misunderstand the use of PSL brackets. Some investigators have assessed the rates of space closure and canine retraction for PSL and CV brackets and have found no statistical differences for the appliance types [29,30]. They have described the treatment procedure generally as follows: After the initial alignment, an SS wire was placed for canine retraction or for space closure. Therefore, canine retraction or space closure takes several months after the appliance is bonded. If the leveling process takes several months, the RAP ends in the meantime.

We use PSL brackets to eliminate the force of ligation and to advance the leveling of the anterior teeth before the RAP ends. If the canine teeth are moved distally from the beginning of treatment, the wire sequence can be advanced at almost every visit, and a working wire can be placed in a few months.

### 2.4. Bi-Dimensional Slots

For a space closure through the use of sliding mechanics [24], the frictional and binding force applied to the brackets placed in the molar region should be low. However, when a larger archwire, such as a working wire (0.016 × 0.0022 inch), is used with an increase in friction and binding [12], a smooth space closure becomes more difficult. In contrast, with a smaller archwire, the torque is lost in the anterior region, and lingual inclination occurs. Thus, two apparently contradictory actions must occur at one time: providing sufficient torque in the anterior region and reducing the friction and binding in the molar region to close the extraction space through the application of a reduced force. When we started using the JET system, we used the same-size brackets throughout and SS wires with rectangular anterior and round molars, but the results were not as good as they could have been because wires were available in very few sizes and no wires were made of Ni-Ti.

The use of anterior brackets with varying slot sizes allows for a wide selection of wires of many sizes and made of different materials. We use anterior brackets that have 0.018-inch slots and posterior brackets that have 0.022-inch slots.

### 2.5. Rapid Wire Sequence

RAP-induced tooth movement is only effective during the first four months after tooth extraction [31]. Therefore, it is crucial to have as much movement of the canine teeth as possible while the socket is still active for a successful reduction of the overall treatment duration.

Frictional and binding forces with CV brackets have been reported to be 100 gf (3.5 ozf) [32]. When a canine retraction is performed with a CV bracket and a 50 gf (1.8 ozf) Ni-Ti closed coil spring on a 0.014 inch heat-activated Ni-Ti (HANT) wire, the frictional force exceeds the force of the canine retraction, and so the canine teeth do not move over the wire, and the wire and the canine teeth are tilted distally in unison. In contrast, canine retraction with PSL brackets and a 50 gf (1.8 ozf) Ni-Ti closed coil spring on a 0.014-inch HANT wire eliminates crowding as the canine teeth slide over the wire. Although some distal tipping occurs with these materials, the distal movement of the canine teeth results in increased leveling, and in many cases the wire can be replaced with a 0.016 × 0.016 HANT wire the next month, which in turn can be replaced with a 0.017 × 0.022 inch Ni-Ti wire in the following month. Because of the rapid wire changes, distal tipping of the canine does not occur.

### 2.6. Initial Elastic (Intermaxillary Elastic)

To start the finishing step of the treatment, we insert intermaxillary elastics immediately after the orthodontic appliance is placed. The major difference between the traditional method and JET system with regard to the use of intermaxillary elastics is the amount of force applied. Conventionally, it is common to apply a force exceeding the ligation force (approximately 100 gf (3.5 ozf)) to the intermaxillary elastics; with the JET system, a force of only 50 gf (1.8 ozf) is necessary.

### 2.7. Indirect Bonding

Joiner [33] reported that bracket positioning errors in indirect bonding were much less common than those in direct bonding. We also use indirect bonding to shorten the treatment period. Currently, indirect bonding is the only method for the accurate intraoral placement of bracket positioning on an ideal setup model. Although a wide variety of indirect bonding trays are currently available, those made of materials easy to process and capable of a consistent application are preferable. We currently use silicone trays whose occlusal surfaces are reinforced with resin for all teeth.

### 2.8. Oral Myofunctional Therapy (MFT)

In the JET system, the LCF is used; therefore, tongue thrust upon swallowing and pronunciation and weak lip-closing pressure have greater effects on the treatment outcome with the JET system than with the conventional procedure. A stagnation of the space closure and occlusion’s failure to stabilize at the final step can delay the removal of the orthodontic appliance. Myofunctional therapy is essential for the smooth progress of treatment.

## 3. Case Presentation

### 3.1. Diagnosis and Etiology

A 15 year old Japanese girl presented with a chief complaint of infralabioversion of the canine teeth in the maxillary arch and irregularly aligned anterior teeth in both arches. Her medical history included no allergies or medical problems, and she showed no signs or symptoms of temporomandibular dysfunction.

Pretreatment facial photographs revealed a convex facial profile and slight facial asymmetry, but no problems with molar relationships. In comparison with her facial midline, the maxillary dental midline was shifted 1 mm to the right. The patient had Angle Class I malocclusion with a 1.0 mm overjet and a 1.0 mm overbite. Both the maxillary and mandibular arches were irregularly aligned, with a −5.0 mm discrepancy in the maxillary arch length and a −3.5 mm discrepancy in the mandibular arch length (Figure 2). A panoramic radiograph showed that none of her third molars had erupted. The lateral cephalometric analysis indicated a normal skeletal relationship with an ANB angle of 4.7°, a slight high Frankfort mandibular plane angle (FMA) of 32.6°, and labially inclined incisors with the maxillary central incisor at a Frankfort plane angle (FH-U1) of 113.5° and a mandibular central incisor at a Frankfort mandibular incisor angle (FMIA) of 44.2° (Table 1). This diagnosis with Angle Class I malocclusion was with a bimaxillary protrusion.

### 3.2. Treatment and Objectives

The main objective of treating the malocclusion was to improve the infralabioversion of the maxillary canine teeth, to alleviate crowding of the mandibular lateral incisors, and to correct the labial inclination of the maxillary incisors and the mandibular lip protrusion.

### 3.3. Treatment Alternatives

In general, to reduce crowding and improve maxillary protrusion, extraction of the four first premolars is the first option to be considered. The second option is to extract the maxillary first premolars and mandibular second premolars if there is a large amount of mesial movement of the mandibular first molars. The third choice is to only extract the maxillary first premolar and perform a distal movement of the mandibular arch. When the mandibular second premolars were extracted, the mesial movement of the mandibular first molar was about 2.8 mm. We chose to extract the mandibular second premolars as well because the distal movement of the mandible would be larger if the teeth were not extracted. Since the molar relationship was Class I, the maxilla was moderately anchored. For the mandible, we decided to extract the second premolar and move the first molar proximally. However, if the molar relationship was to be disrupted, orthodontic anchor screws were to be placed.

### 3.4. Treatment Progress

We planned treatment by the JET system using PSL brackets (Clarity^TM^ SL and SmartClip^TM^ Self-Ligating Brackets, 3M Unitek, Monrovia, CA, USA) of varying slot sizes: a 0.018-inch slot for anterior brackets (3-3) and a 0.022-inch slot for posterior brackets. This protocol was chosen to prevent the lingual transposition of the maxillary teeth and alleviate the mandibular right and left premolars; the mandibular right and left second premolars were extracted, and the orthodontic appliance was placed the next day.

Brackets were positioned on a plaster model in advance (straight-wire positioning); thereafter, they were bonded using an indirect tray. The maxillary right and left canine distal movement and mandibular right and left first molars were initiated immediately after placing the brackets using 50 gf (1.8 ozf) Ni-Ti closed coil springs. In addition, 1/4-inch intermaxillary elastics were placed between the maxillary canines and mandibular first premolars immediately after bracket placement to avoid mesial tipping of the mandibular first molars. At this point, the maxillary and mandibular anterior retraction were initiated using 100 gf (3.5 ozf) Ni-Ti closed coil springs, and brackets were placed on the second molars four months after the initial placement. The extraction space was closed seven months after bracket placement.

Although the detailing of the treatment took some time, the total duration of the treatment was 13 months. Both the upper and lower lips recessed, and the patient’s facial profile was considerably improved. The postoperative panoramic radiograph identified a slight root resorption and no problem with root parallelism. The postoperative cephalometric analysis in the lateral view demonstrated changes in the FMIA (44.2 to 57.2) and interincisal angle (110.7 to 136.1), which contributed to an improvement in the facial profile.

After active treatment, the pre-adjusted edgewise appliance was removed. The patient wore a circumferential retainer on both arches full-time for the first year and then only at night.

### 3.5. Treatment Results

The patient showed an acceptable occlusion and good facial profile (i.e., balanced lip line), owing to the successful retraction of the upper canines and anterior teeth. The dental arches were aligned and leveled, and an ideal overjet and overbite were achieved (Figure 3).

During the active treatment, at three years and six months after debonding, no significant periodontal problems, such as gingival recession or loss of tooth vitality, were observed. Panoramic radiographs before and after treatment showed no significant reduction in the crest bone height and a slight root resorption (Figure 4).

Cephalometric superimpositions before and after the treatment showed a slight mesial movement of the maxillary molars. Owing to there being no serious anchorage loss, the Class I molar relationship was maintained (Figure 5).

## 4. Discussion

We have for the first time described a novel technique for the fixed orthodontic treatment of moderately crowded teeth. This technique, the JET system, yielded predictable outcomes, and the orthodontic treatment was completed in a relatively short time (13 months).

Angle Class I molar and canine relationships were established at the end of the treatment. Moreover, crowding in the mandible and maxilla was corrected, and an optimal overlap of the upper incisors was established. No scar tissue was observed in any gingival region.

In the JET system, extraction causes the RAP, and a 50 gf (1.8 ozf) NITi closed coil is used to move the canine tooth distally with a 0.014 HANT wire in the main arch. Because we used PSL brackets, we could place a 0.016 × 0.016 HANT wire after one month. One month later, 0.017 × 0.022 Ni-Ti wires were placed and retained until the end.

Darendeliler et al. [34] reported that light force (50 gf) achieved the full space closure without distal tipping or rotation of the canine and mesial tipping or rotation of the molars. Furthermore, light force (50 gf) provided a greater percentage of canine retraction than heavy force (300 gf) without anchorage loss [17]. Moreover, there was no difference in the anchorage loss between SL brackets and CV brackets during canine retraction [35]. Therefore, with the JET system with the RAP, the LCF and LF, no distal tipping might occur during canine retraction (Figure 6).

One of the features in the JET system is the movement of teeth immediately after extraction. The condition of the tissue surrounding the sockets is changed by the RAP. According to Verna [36], the RAP occurs during the healing process of the alveolar sockets following tooth extraction and during the orthodontic tooth movement in the alveolar bone. Therefore, the JET system might be accelerated by an increased bone metabolism in the RAP.

These reports suggest that the JET system could stimulate bone synthesis and bone resorption and accelerate tooth movement. Therefore, a combination of the LCF and LF during the RAP can also accelerate tooth movement, modulate the state of the bone metabolism, and activate osteogenesis and osteoclasts, which enabled us to finish the active treatment in 13 months.

## 5. Conclusions

The JET system, which entails the use of the LCF, LF and RAP, might shorten the orthodontic treatment period (to 13 months in our patient) without serious anchorage loss or adverse effects. At follow-up three years and six months later, the stability of the treatment in our patient was excellent.

## Figures and Tables

**Figure 1 medicina-58-00150-f001:**
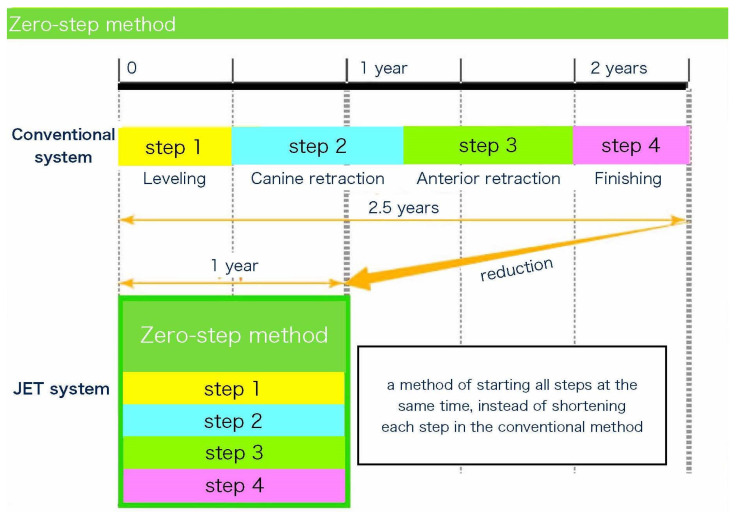
Comparison of conventional orthodontic treatment with the Jiyugaoka Enjoyable Treatment (JET) system.

**Figure 2 medicina-58-00150-f002:**
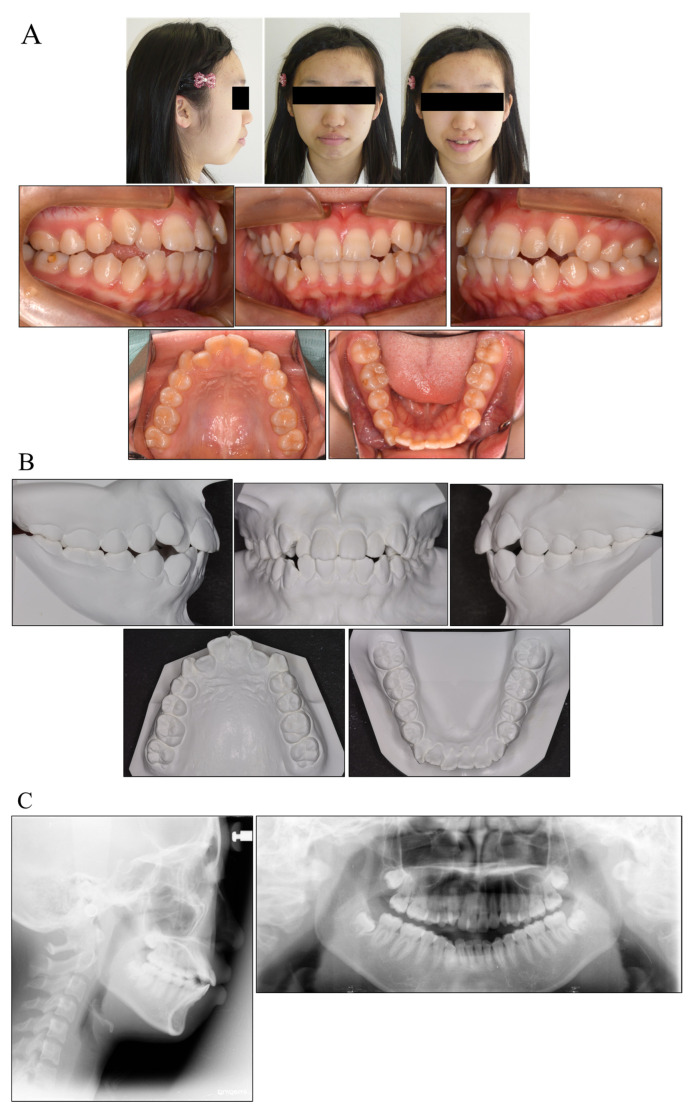
Facial and intraoral photograph (**A**), dental casts (**B**) and radiographs ((**C**) lateral cephalogram; panoramic radiograph) of pretreatment (patient is 15 years old).

**Figure 3 medicina-58-00150-f003:**
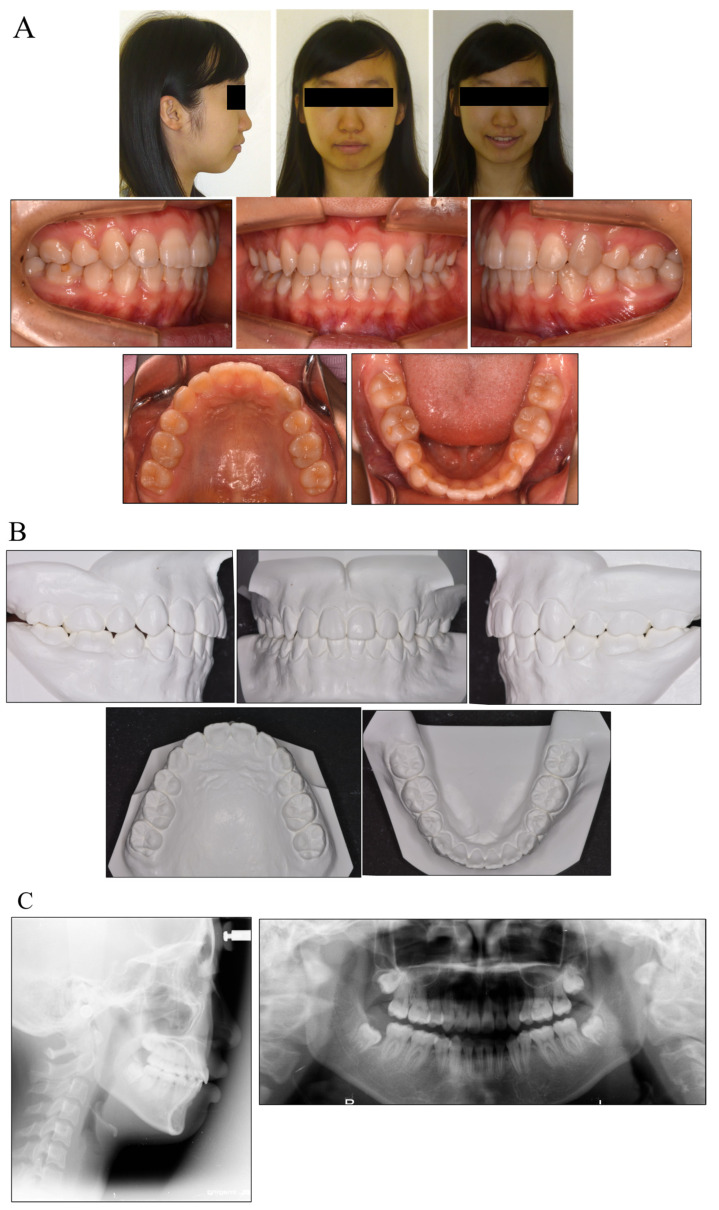
Facial and intraoral photograph (**A**), dental casts (**B**) and radiographs ((**C**) lateral cephalogram; panoramic radiograph) of post-treatment (patient is 17 years old).

**Figure 4 medicina-58-00150-f004:**
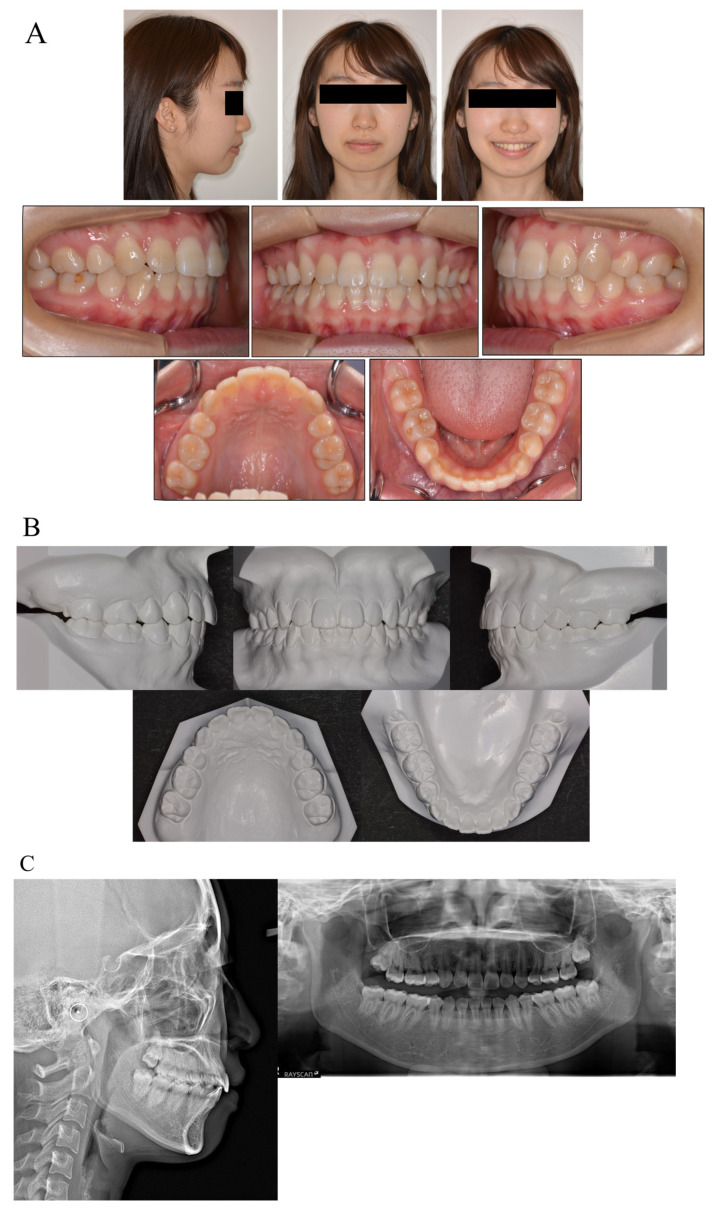
Facial and intraoral photograph (**A**), dental casts (**B**) and radiographs ((**C**) lateral cephalogram; panoramic radiograph) of post-treatment at three years and six months (patient is 20 years old).

**Figure 5 medicina-58-00150-f005:**
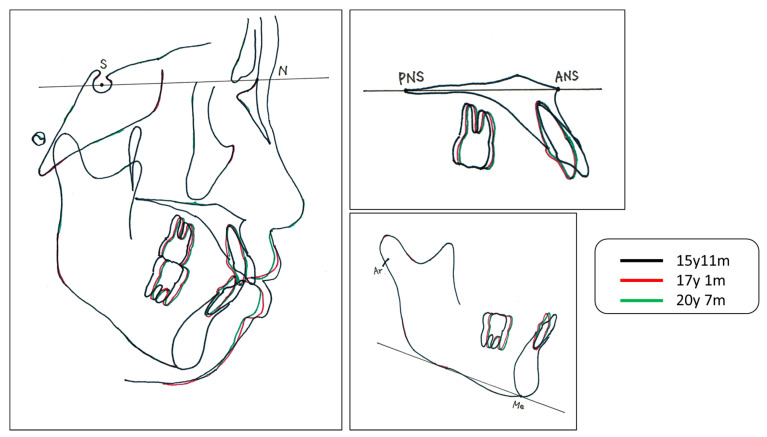
Cephalogram tracing. S; Sera turcica, N; Nasion, PNS; Posterior nasal spine, ANS; Anterior nasal spine, Ar; Articulare, Me; Menton.

**Figure 6 medicina-58-00150-f006:**
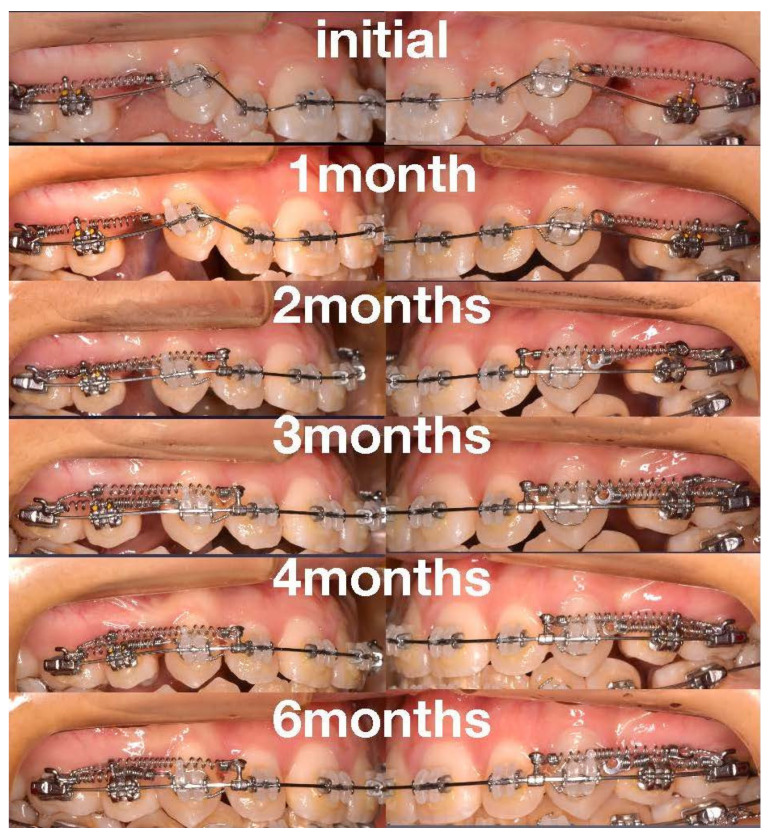
Image of no distal tipping during canine retraction.

**Table 1 medicina-58-00150-t001:** Cephalometric measurement at pretreatment and posttreatment.

Variable	Japanese Norm (±S.D.)	Pretreatment (15y 11m)	Posttreatment (17y 1m)	3 Years 6 Months after Debonding (20y 7m)
SNA (°)	81.3 ± 3.5	81.6	81.6	81.6
SNB (°)	78.9 ± 3.5	76.9	76.9	76.9
ANB (°)	3.4 ± 1.8	4.7	4.7	4.7
FMA (°)	28.8 ± 4.1	32.6	32.6	32.6
FMIA (°)	56.9 ± 6.4	44.2	57.2	56.9
IMPA (°)	96.3 ± 5.8	103.2	90.2	90.5
FH-U1 (°)	111.1 ± 5.5	113.5	101.1	101.8
U1-L1 (°)	124.1 ± 7.6	110.7	136.1	135.1
Gonial angle (°)	122.2 ± 4.6	124.2	124.2	124.2
E-line:Upper (mm)	+2.0 ± 2.0	+2.9	+0.3	−0.8
E-line:Lower (mm)	+2.0 ± 2.0	+6.3	+3.1	−0.1

SNA; The angle between S-N plane and N-A plane, SNB; The angle between S-N plane and N-B plane, ANB; The angle between N-A plane and N-B plane, FMA; The angle between Mandibular plane and F-H plane, FMIA; Lower 1 to F-H plane angle, IMPA; Lower 1 to mandibular plane angle, FH-U1; Upper 1 to FH (Frankfort horizontal) plane angle, U1-L1; The angle between Upper 1 and Lower 1, Gonial angle; The angle between Ramus plane and mandibular plane, E-line: Upper; Upper lip to Esthetic plane, E-line: Lower; Lower lip to Esthetic plane.

## Data Availability

Not applicable.

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
