# Peer review of "A Novel Technique for Shortening Orthodontic Treatment: The “JET System”"

_medicina, 2022, doi:10.3390/medicina58020150_

Round 1
Reviewer 1 Report
General comment: The authors tackled a new and interesting orthodontic topic! But, being a particular and useful technique for orthodontic specialists, the authors needed to better describe their work and not get confused. Starting with the title, it's a bit confusing for the reader. The referee does not agree with the term "system", but suggests calling it "technique". In the abstract it is said that "the patient's facial appearance was symmetrical", but looking at the photographs she had an asymmetrical face, because in this orthodontic diagnosis it is impossible to have a symmetry, due to the anterior crowding, the cross bite and tendency to unilateral open bite, which in this case can be noticed on the right.
The authors cannot start from the intraoral situation, but describing the diagnosis before treatment by putting the cephalometric data, the OPT before starting, the intra and extraoral photos and then continuing for each month and year the trend of the new technique.
There are several incorrect technical / scientific terms, which the authors need to deepen this topic on the basis of several scientific articles.
With regard to root resorption, the authors cannot say that no root resorption has been found, because even with the naked eye the difference between one OPT and the other is observed and this must be verified by measuring each root length before and after treatment and even over the years. The main goal for an orthodontist is to keep root resorption under control and this is inevitable in orthodontics; in this way the authors could mention as the main objective for their new technique to be achieved at the end of the treatment. The authors must base their status on scientific papers for treatments with more than one year of treatment.
Each of the sections needs a new reformulation and organization following the chronology. While reading you can find several repeated sentences/phrases. As a manuscript is very poor and described not with a scientific quality, where English is with many grammatical errors.
Author Response
Manuscript ID: medicina-1489823 - Major Revisions
Response to Reviewer 1 Comments
Thank you for giving us valuable advice and comments regarding our manuscript. We revise upon the manuscript as follows.
Reviewer 1
Point 1:
General comment: The authors tackled a new and interesting orthodontic topic! But, being a particular and useful technique for orthodontic specialists, the authors needed to better describe their work and not get confused. Starting with the title, it's a bit confusing for the reader. The referee does not agree with the term "system", but suggests calling it "technique".
Response 1: We are honored by your kind words.
As you advised, the title may be a bit confusing for the reader.
The JETsystem was developed to shorten the duration of orthodontic treatment in cases of a tooth extraction with light continuous force (LCF) and low friction (LF) while utilizing the RAP condition. (P.2, L. 65-66.)
Therefore, we have simply changed the title to “ A new technique called the “JETsystem”
Point 2:
In the abstract it is said that "the patient's facial appearance was symmetrical", but looking at the photographs she had an asymmetrical face, because in this orthodontic diagnosis it is impossible to have a symmetry, due to the anterior crowding, the cross bite and tendency to unilateral open bite, which in this case can be noticed on the right.
Response 2: This case has passed the examination of the Board Certified Clinical Trainer of the Japanese Orthodontic Society.
In that review, there were no special remarks after diagnosing it as symmetry, so we believe there is no problem.
Point 3:
The authors cannot start from the intraoral situation, but describing the diagnosis before treatment by putting the cephalometric data, the OPT before starting, the intra and extraoral photos and then continuing for each month and year the trend of the new technique.
There are several incorrect technical / scientific terms, which the authors need to deepen this topic on the basis of several scientific articles.
Response 3: Your opinion is justified. However, we want to emphasize the concept of new technique by showing the flow of treatment. Therefore, the intra and extraoral photos and then continuing for each month and year were presented in this case report.
References have been added and some sentences have been revised.
Point 4:
With regard to root resorption, the authors cannot say that no root resorption has been found, because even with the naked eye the difference between one OPT and the other is observed and this must be verified by measuring each root length before and after treatment and even over the years. The main goal for an orthodontist is to keep root resorption under control and this is inevitable in orthodontics; in this way the authors could mention as the main objective for their new technique to be achieved at the end of the treatment. The authors must base their status on scientific papers for treatments with more than one year of treatment.
Response 4: As you advised, root resorption was slightly observed, so the description was rewritten.
- 8, L. 222.
Point 5:
Each of the sections needs a new reformulation and organization following the chronology. While reading you can find several repeated sentences/phrases. As a manuscript is very poor and described not with a scientific quality, where English is with many grammatical errors.
Response 5: As you indicated, this text was received professional English editing by MDPI English editing. Some sentences have been revised.

Reviewer 2 Report
Dear Authors,
The study is of scientific interest and in line with the aims of the journal. The author guidelines have been respected, the work is well written, and the case report was well documented.
However, there are some issues that should be addressed, such as the lack of references in the Introduction Section that should be largely improved. Moreover, Introduction Section should be improved highlighting the clinical features that could lead to choosing the extractive treatment.
Minor revisions:
Abstract:
- “using by”
- “light continuous force” should be “forces”. Please modify in all the text.
- Introduction
- “Factors, such as patient age, malocclusion type, extractions presence/ absence, techniques applied by means of fixed appliances, ligation method are involved”. Please add reference
- “In recent years, a series of methods have been proposed to limit the frictional restraints that contrast tooth movement at the bracket archwire-ligature interface, such as SLBs (SLBs) and unconventional ligature system”. Please add reference
- “Among SLBs, those in which the self- ligating clip does not press against the archwire (passive SLBs) have consistently shown a smaller amount of friction than the conventional system. Therefore, the possibility that this bracket system may influence treatment duration has been advanced”. Please add reference
- Please add a section explaining the conditions under which an extraction treatment is chosen (e.g. crowding, including canine, open bite). Please, add this reference “Bizzarro M, Generali C, Maietta S, Martorelli M, Ferrillo M, Flores-Mir C, Perillo L. Association between 3D palatal morphology and upper arch dimensions in buccally displaced maxillary canines early in mixed dentition. Eur J Orthod. 2018 Nov 30;40(6):592-596. doi: 10.1093/ejo/cjy023.”
- “In previous studies, comparisons were made at a time when structural differences in the brackets did not affect treatment duration”. Please add reference
- Concepts of JETsystem - 2.1. Passive self-ligating brackets
- “In order to maximize the use of the immediate post-extraction the RAP status for tooth 84 movement”. Please rephrase, it is not clear.
- Treatment alternatives
- Could you explain better why do you chose the extraction of the first premolars in the maxilla and the second premolars in the mandible?
- Treatment progress
- Why did you use 1/4 inch intermaxillary elastics between the maxillary canines and the mandibular first premolar? Why first premolar e not first molar?
Author Response
Manuscript ID: medicina-1489823 - Major Revisions
Response to Reviewer 2 Comments
Thank you for giving us valuable advice and comments regarding our manuscript. We revise upon the manuscript as follows.
Reviewer 2
The topic is of interest as inflammatory root resorption occurs frequently during and after orthodontic therapy.
Response: We are honored by your kind words.
Minor revisions:
Point 1:
Abstract:
“light continuous force” should be “forces”. Please modify in all the text.
Response 1: As you advised, “light continuous force” was changed to “forces” and “LCF” in the text.
Point 2:
Introduction
“Factors, such as patient age, malocclusion type, extractions presence/ absence, techniques applied by means of fixed appliances, ligation method are involved”. Please add reference
Response 2: As you advised, we revised this sentence as follows:
Factors, such as patient’s cooperation [2-5], malocclusion severity [3-5], extractions presence/ absence [3-7], techniques applied by means of fixed appliances [5,8], bracket prescription [9] are involved.
- 1, L.32-37.
Point 3:
“In recent years, a series of methods have been proposed to limit the frictional restraints that contrast tooth movement at the bracket archwire-ligature interface, such as SLBs (SLBs) and unconventional ligature system”. Please add reference
Response 3: As you advised, we revised this sentence as follows:
Self-ligating brackets (SL) and low friction (LF) ligated brackets [11] have long been proposed as a way to release the frictional resistances that control orthodontic tooth movement (OTM) at the bracket-archwire-ligature interface. Burrow [12] reports that SL do not shorten treatment time because the effect of binding and notching, not friction, is significant, and the binding-and-releasing phenomenon occurs in SL as well as in con-ventional brackets. Among SL, those in which the self-ligating clip does not press against the archwire called passive Self-ligating brackets (PLBs) [13].
- 1, L. 38-44.
Point 4:
“Among SLBs, those in which the self- ligating clip does not press against the archwire (passive SLBs) have consistently shown a smaller amount of friction than the conventional system. Therefore, the possibility that this bracket system may influence treatment duration has been advanced”. Please add reference
Response 4: As you advised, we added two references as follows:
Among SLBs, those in which the self-ligating clip does not press against the archwire (passive SLBs) [13] have consistently shown a smaller amount of friction than the conventional system [14]. Therefore, the possibility that this bracket system may influence treatment duration has been advanced.
- 1, L. 43- P. 2, L 47.
Point 5:
Please add a section explaining the conditions under which an extraction treatment is chosen (e.g. crowding, including canine, open bite). Please, add this reference “Bizzarro M, Generali C, Maietta S, Martorelli M, Ferrillo M, Flores-Mir C, Perillo L. Association between 3D palatal morphology and upper arch dimensions in buccally displaced maxillary canines early in mixed dentition. Eur J Orthod. 2018 Nov 30;40(6):592-596. doi: 10.1093/ejo/cjy023.”
Response 5: As you advised, we added a sentence as follows:
Japanese people often have a face with a bimaxillary protrusion. In addition, when the maxillary canines are in a high labial dislocation, the dental arch in the maxilla be-comes narrower [10], and extraction is often needed.
- 1, L.35-37.
Point 6:
“In previous studies, comparisons were made at a time when structural differences in the brackets did not affect treatment duration”. Please add reference
Response 6: As you advised, we added two references as follows:
In previous studies [22,23], comparisons were made at a time when structural differences in the brackets did not affect treatment duration.
- 2, L.59-61.
Point 7:
Concepts of JETsystem - 2.1. Passive self-ligating brackets
“In order to maximize the use of the immediate post-extraction the RAP status for tooth 84 movement”. Please rephrase, it is not clear.
Response 7: As you advised, the sentence was revised.
To make the most of the RAP state immediately after extraction for tooth movement,
- 4, L. 83-86.
Point 8:
Treatment alternatives
Could you explain better why do you chose the extraction of the first premolars in the maxilla and the second premolars in the mandible?
Response 8: As you advised, the sentence was revised.
We chose to extract the mandible as well because the distal movement of the mandible would be larger if the teeth were not extracted. Since the molar relationship was class I, the maxilla was moderately anchored. For the mandible, due to the large amount of molar mesial movement, we decided to extract the second premolar and move the first molar mesially.
- 7, L. 192-193.
Point 9:
Treatment progress
Why did you use 1/4 inch intermaxillary elastics between the maxillary canines and the mandibular first premolar? Why first premolar e not first molar?
The system that combines these advantages (low friction and light continuous forces) was introduced by Damon [19]. As per the proponents of this system, the low-forces and low-friction environment provided by the Damon appliance offers considerable advantages over those with conventional ligation.
Response 9: As you advised, the sentence was revised.
In addition, a 1/4-inch intermaxillary elastics was placed between the maxillary canines and mandibular first premolars immediately after bracket placement to avoid mesial tipping of the mandibular first molars.
- 8, L. 213-216.

Round 2
Reviewer 1 Report
General comment: Thank you to the authors for their changes and responses! But the referee again has noticed that the English has a lot of mistakes, the verb tenses are still confused; the authors start a sentence in the present tense and the following one in the past one. The text has not been changed extensively, but only a few sentences. The referee cannot judge the colleagues about the facial asymmetry, but an orthodontist specialist can notice well the asymmetry in this patient presented. In the Introduction section, Figure 1 makes no sense where the authors have put the figure. They must insert the figures in each subsection where they mention the respective figure and continuing with the description of the text and so on. In the text the retainer (Line 227) is mentioned, but in the photos shown at the end it is not observed. The retainer has always been the most important step at the end of the orthodontic treatment, also mentioning the failures. Where they write “Figs. 2 A, B, C ”, in the figure the letters are not present and in the figure legend they are not described. This is also evident for the other figures. Figure 6 is very poor and the authors must also put the right, not just the left. Figure 6 in point 7 is not cited and it makes no sense here after the figures, which thus presented create confusion again. In the end, photos after 3.6 years are necessary to be put as a follow-up and not in the introduction section. Please, convert gf to oz!
Author Response
Manuscript ID: medicina-1489823 - Major Revisions
Response to Reviewer 1 Comments
Thank you for giving us valuable advice and comments regarding our manuscript. We revise upon the manuscript as follows.
Reviewer 1
Point 1:
General comment: Thank you to the authors for their changes and responses! But the referee again has noticed that the English has a lot of mistakes, the verb tenses are still confused; the authors start a sentence in the present tense and the following one in the past one. The text has not been changed extensively, but only a few sentences.
Response 1:
As you indicated, this text was received professional English editing by MDPI English editing. Furthermore, it was received another professional English editing by native english editing.
Point 2:
The referee cannot judge the colleagues about the facial asymmetry, but an orthodontist specialist can notice well the asymmetry in this patient presented.
Response 2:
As you indicated, the patient presented slight facial asymmetry, but it is considered that there is no therapeutic problem.
- 5, L. 191-192.
Point 3:
In the Introduction section, Figure 1 makes no sense where the authors have put the figure. They must insert the figures in each subsection where they mention the respective figure and continuing with the description of the text and so on.
Response 3:
As you indicated, Fig. 1 was changed to a schema because it was difficult for the reader to understand and was thought to be confusing.
Point 4:
In the text the retainer (Line 227) is mentioned, but in the photos shown at the end it is not observed. The retainer has always been the most important step at the end of the orthodontic treatment, also mentioning the failures.
Response 4:
As you indicated, the retainer is very important in orthodontic treatment. However, generally, We don't think photos with retainers are shown.
The post-treatment and the 3 years and 6months follow-up after post-treatment are shown in Figs. 2 and 3, respectively.
Point 5:
Where they write “Figs. 2 A, B, C”, in the figure the letters are not present and in the figure legend they are not described. This is also evident for the other figures.
Response 5:
As you indicated, “A, B, C” were deleted in Figs. 2, 3, and 4.
Point 6:
Figure 6 is very poor and the authors must also put the right, not just the left. Figure 6 in point 7 is not cited and it makes no sense here after the figures, which thus presented create confusion again.
Response 6:
As you indicated, the figures of right side added to Fig. 6. We added the references about tipping of canine and anchorage loss as follows;
Darendeliler et al. [34] reported that light force (50-gf) archived the full space clo-sure without distal tipping or rotation of canine and mesial tipping or rotation of the molars. Furthermore, light force (50-gf) provided a greater percentage of canine retrac-tion than heavy force (300-gf) without anchorage loss [17]. Moreover, there was no difference in anchorage loss between SLBs and conventional brackets during canine retraction [35]. Therefore, JET system with the regional acceleratory phenomenon (RAP), light continuous forces (LCF) and low friction might occur no distal tipping during canine retraction (Figure 6).
- 11, L. 293-299.
Point 7:
In the end, photos after 3.6 years are necessary to be put as a follow-up and not in the introduction section. Please, convert gf to oz!
Response 7:
The 3 years and 6months follow-up after post-treatment are shown in 3, respectively.
The ounce notation was added.

Round 3
Reviewer 1 Report
The authors have corrected and improved their manuscript basing in the comments made by the referee!